# Electric-Contact Tilt Sensors: A Review

**DOI:** 10.3390/s21041097

**Published:** 2021-02-05

**Authors:** Sergiusz Łuczak, Magdalena Ekwińska

**Affiliations:** Warsaw University of Technology, Faculty of Mechatronics, 02-525 Warsaw, Poland; magdalena.ekwinska@pw.edu.pl

**Keywords:** tilt sensor, inclinometer, Platonic solid, mercury sensor, accelerometer

## Abstract

A review of various kinds of solid tilts sensors, using a free mechanical member for generation of electric-contact (mostly a ball), is presented. Standard and original solutions are discussed. The latest patents are described. A classification of the existing solutions with respect to their sensing principle is proposed. Possible types of the electric/electronic circuits are discussed. Advantages of these sensors are emphasized: mainly optional operation without power supply, resistance to electrostatic discharges, and simplicity of signal processing. Technological details are briefly introduced, along with miniaturization prospects. Additionally, liquid tilt sensors are succinctly characterized. The most typical tilt sensing techniques are concisely compared.

## 1. Introduction

Tilt is a mechanical quantity—a specific kind of angular displacement, generally expressed by two components: pitch and roll. It appears in many situations due to the law of universal gravitation. As far as human sensation is concerned, an interesting fact is that the sense of balance (equilibrium) is one of the basic six senses using external receptors [1]. Even more startling is a similarity between this sense and some of mechanical tilt sensors presented later in the text, using a free member for generation of electric-contact between the sensing electrodes and featuring full detection range. External receptors of the human sense of balance consist not only of the semicircular canals but also of the otolith system, which includes the utricle and the saccule [2], and is located next to the semicircular canals and contains numerous otoliths (crystals of calcium carbonate). The otoliths can freely move while the human accelerates or tilts its head, and thus represent linear motion sensors [2]. Even though the otoliths do not generate an electric contact, as the discussed tilt sensors, they stimulate appropriate receptor cells while relocated or accelerated. Therefore, some sensors presented in the text resemble, in a way, the sense of balance not only of humans (and other mammals), but even of fish, whose otoliths are used as indicators of gravity, balance, movement, and orientation [3].

Despite being a very old solution, tilt sensors with mechanical elements creating electric contact are still used in various branches of industry, e.g., extractive industry while drilling holes for pipelines [4]. These sensors feature such advantages as robustness, simple structure, simplicity of the signal processing (no necessity of calibration), possibility of operation without a power supply, small overall dimensions/low mass, high durability, and reliability. They can operate only under static conditions, since vibration and external acceleration completely disturb their operation.

There exist more sophisticated versions of tilt sensors with free active mechanical elements, e.g., [5,6,7,8], which employ certain physical phenomenon to generate analog signals. Another solution employed for tilt sensing, interesting from the point of view of high sensitivity and simplicity, is application of resonance phenomena, presented, e.g., in [9]. However, in the opinion of the authors, such sensors cannot compete with accelerometers—especially manufactured in micro-electro mechanical systems (MEMS) technology or in a conventional technology using, e.g., strain gauges [10], which are often used for tilt measurements [11]. Fundamentals of employing MEMS accelerometers for tilt measurements as well as evaluation of their accuracy (being typically in the order of 0.3°) are reported in [12], whereas some novel measurement methods can be found, e.g., in [13].

When not only an accelerometer but also a gyroscope is employed (often contained within a single package), it works even more in favor of microsensors. In this way, a dynamic inclinometer is created [14], which can operate under static, quasi-static, and dynamic conditions.

Then, the scope of applications of MEMS inertial sensors in tilt measurements becomes much wider. The most typical as well as specific applications have been listed in [15]. The interesting fact is that the number of novel devices employing tilt measurements constantly increases, including, e.g., self-optimizing orthotic robots [16] or new generation of motorcycle anti-lock braking system (ABS) [17]. It seems that in some of these applications, the considered mechanical tilt sensors could be applied as successfully as MEMS devices.

Despite the advantages of the aforementioned MEMS devices, simple electric contact tilt sensors, even though they feature much worse operational parameters, have still been in use and have their place in tilt measurements and detection. Being aware of their many different designs minutely discussed in the text (including the biomimetic ones), it is possible to purchase or design a sensor that suits well a specific application.

The review is organized in the following way: Section 2 briefly describes liquid tilt sensors; Section 3 presents tilt switches, mostly commercial; Section 4 describes tilt sensors featuring full detection range, divided both into non-uniform as well as uniform sub-ranges; Section 5 discusses the relevant electronic circuits and possible ways of generating and processing the output signals; Section 6 reports on experimentally determined operational parameters of selected sensors; Section 7 addresses some manufacturing issues; Section 8 summarizes the most important findings and compares operational parameters of the most typical types of tilt sensors; Section 9 describes prospective development of electric contact tilt sensors in the nearest future; Section 10 considers shortcomings and advantages of electric contact tilt sensors to indicate their best scope of application.

## 2. Liquid Tilt Sensors

The most common type of tilt sensor are liquid inclinometers—spirit (alcohol) levels, first of all. They are widespread and used all over the world, serving mainly as a leveling instrument for such craftsmen as carpenter and mason, as well as for ordinary people (e.g., leveling of scales in grocery stores).

As far as liquid tilt sensors with electric output are concerned, usually mercury or an electrolyte is utilized as the conductive medium. Only a part of the sensor volume may be filled with the liquid, leaving some space for a non-conductive fluid (a liquid of high electric resistance, a volume of gas, or a small air bubble). Due to its toxicity, mercury sensors are rather withdrawn from the market, however have been manufactured yet and still operate in many obsolete devices (e.g., old types of air-condition thermostats still used in the United Sates—see Figure 1a at the furthest right). Moreover, some designs have been developed for research purposes quite recently, e.g., for monitoring of animal movements [18]. Some researchers proposed even a modern novel design of a mercury tilt sensor manufactured using MEMS technology [19] in the last decade.

The principle of operation of the liquid contact sensors can be basically divided as follows:short-circuiting of two electrodes (see Figure 1a,b),short-circuiting of more than two electrodes,short-circuiting of all electrodes but one, andchanging the length of the part of a resistive element, which is immersed in the liquid (see Figure 1c, note that the resistive wire inside the vial is hardly visible).

These sensors usually operate as simple switches and are tilted only around one axis. However, there exist also dual-axis designs that provide information about both pitch and roll, e.g., electrolytic sensors used in old aviation instruments [20].

Since liquid tilt sensors are capable of featuring a very high sensitivity, in the order of few arc seconds (or tens of microradians) [21,22] or even below arc second [23], many types of electrolytic inclinometers are still manufactured. Such devices are not contact sensors generating discrete signals, but rather feature analog outputs based on changes of electric parameters of the sensor: mostly resistance or capacity [24]. For instance, resistive sensors are presented in [22,25], capacitive sensors in [26,27]; moreover, there is a whole group of sensors employing various kinds of liquids featuring specific magnetic properties, presented e.g., in [28,29,30].

Still another group of liquid tilt sensors employs various optical techniques for detecting tilt, e.g., recording by cameras position of air bubbles [31], application of fiber Bragg gratings [32,33], laser autocollimation [34], application of interferometer [35,36], application of optical fiber [37], or application of standard photodetectors illuminated by a light source [38,39].

Even though some attempts have been made to miniaturize liquid sensors, reported, e.g., in [23,40], the volume of the liquid must be at least few tens of mm^3^, otherwise huge mechanical hysteresis cannot be avoided, due to high adhesion forces and surface tension [41].

## 3. Tilt Switches

Tilt switches usually detect only horizontal position within a certain angular range (i.e., vicinity of the horizontal position), typically of few tens of arc degrees. Often, they do not operate within the full solid angle and generate no electric contact between the electrodes, while oriented beyond the angular range deemed the horizontal position, or just the opposite: no electric contact while in horizontal position. As far as the graphical illustrations of the sensors presented later in the text are concerned, it was accepted that the main view corresponds to a horizontal position of the sensor, whereas illustration explaining the principle of operation refers to a case when the sensor is tilted and generates the output signal.

### 3.1. Commercial Switches

There is a wide variety of commercial tilt contact switches, manufactured by such companies as Alps Electric, OncQue, Panasonic Industrial Devices, C&K, NKK Switches, and Omron Electronics. Their typical overall dimensions are in the range of 2–30 mm, and the tilt ranges of 7°–90°.

As an example, a commercial single-axis tilt switch (by an unknown manufacturer), having the free member in the form of ball with diameter of 9 mm and contact wires with diameter of 0.7 mm, is presented in Figure 2a. The switch generates three possible signals, corresponding to inclination to the left (see Figure 2c), the horizontal position (see Figure 2a,b), or inclination to the right. Both the ball and the contact wires are coated with iridium in order to provide low transition resistance between these elements.

Another commercial dual-axis tilt switch is presented in Figure 3. It features four electrodes: two at the bottom and two at the top. The ball can contact at a time one of the lower electrodes and one of the upper ones, creating thus four possible signals corresponding to four different orientations. Its principle of operation is the same as of the switch with motion detector introduced in Section 3 (except for application of the magnetic members), and is illustrated in respective figure presenting a cross-section of the inclined sensor.

### 3.2. Patented Designs

Two designs described in patents [42,43] feature a structure similar to the dual-axis switch presented in Figure 3. The switch illustrated in Figure 4a also has four electrodes, yet there is created a cavity in the center of the bottom and the top of its casing. If the switch is oriented approximately horizontally (either low-side down or up-side down), the metallic ball remains in the cavity and is not in contact with any of the side electrodes, so no signal is generated. Only while tilted, the ball short-circuits two adjacent respective side electrodes. Thus, the switch generates four possible signals, not distinguishing between low-side down or up-side down orientation, similarly to the dual-axis switch presented in Figure 3. In other words, its detection range covers only half of the full solid angle.

The switch illustrated in Figure 4b, proposed in [42], operates in a similar way to the switch in Figure 4a proposed in [43]. However, it has additional two electrodes: upper and lower, which also have a cavity created at the center. Owing to the additional electrodes, it generates eight possible signals (two of the four side-electrodes are short-circuited either with the upper or the lower one). However, when the ball remains near the center of the cavity, whether in the lower or the upper electrode, no signal is generated and thus it is not possible to distinguish the low-side down from the up-side down orientation, just like in the previous cases. This ambiguity could have been eliminated had the upper electrode been flat or convex in the contact region. Nevertheless, the design presented in Figure 4b can operate while oriented up-side down and low-side down.

Whereas commercially available tilt switches are usually very simple, there are some interesting patented solutions, e.g., [44,45]. Two examples are presented in Figure 5 and Figure 6.

Except for the standard way of detection, when the ball short-circuits the electrodes while inclined to the left or to the right, a dynamic state can be detected when the ball moves between the electrodes. For this purpose, a special magnetic sensor is used (e.g., Hall-effect device) that generates a signal related to the proximity of the ball, which must feature magnetic properties. Moreover, there are two passive magnets applied, attracting the ball in order to reduce the sensor mechanical hysteresis [44].

An original dual-axis tilt switch is presented in Figure 6. The novel feature of the switch is application of electrodes, which from the mechanical point of view operate as flat springs or a membrane. The electrodes have additionally special cuts (not visible in Figure 6) that make them even more flexible [45].

Introduction of the flexible electrodes improves the electric contact between them and the ball (due to higher unit pressure). Owing to this fact, the ball diameter may be smaller, which makes it possible to miniaturize the overall dimensions of the switch, however at the cost of higher mechanical hysteresis.

## 4. Tilt Sensors with Full Detection Range

In many applications, an arbitrary orientation of the tilt sensor is possible. Therefore, the detection range of the sensor must cover the full solid angle (steregon). In such cases, various inner shapes of the casing of the tilt sensor can be envisaged. However, the most reasonable solutions should be as symmetrical as possible, otherwise each detection sub-range will be different, and thus it will be difficult to analyze a current orientation of the sensor. This is an important issue, since in the case of the considered electric contact sensors, simplicity of signal processing is one of their few advantages.

As previously mentioned, most of the commercial and the patented tilt sensors do not feature detection range equal to full solid angle. It results from the fact that the majority of the devices operate only in an approximately vertical orientation. Therefore, there is no need to detect orientations, which are not relevant to the correct operation of such devices.

Versions of the tilt sensors featuring the full detection range, equal to full solid angle, must provide a unique output signal at any orientation, or alternatively there may exist only one detection sub-range, where no output signal is generated.

The basic principle of operation of such sensors is the fact that a free element, in the gravity field, can be immobilized when it is supported at three different points. A physical realization of this idea is presented in Figure 7, where a metallic ball is contained within a cube, created out of six perpendicular electrodes. A position of its stable equilibrium, depicted in Figure 7, takes place when the ball is located within one of its eight vertices, creating thus eight detection sub-ranges. Then, the ball short-circuits three adjacent electrodes, which create that vertex.

A position while the ball is unstable (when surface of one of the electrodes is horizontal and the ball is not in contact with any of the side electrodes) is rather theoretical. Nevertheless, it can be completely excluded by introducing a convex shape of the contact surface of the electrodes (see Section 4.3).

The general principle adopted in original patented designs [46,47] is presented in Figure 8. The full solid angle is divided into six detection sub-ranges: three sub-ranges when the metallic ball is contained within the triangular opening of the lower electrode, and another three sub-ranges when the ball is within the triangular opening of the upper electrode. The triangular openings are rotated by 60° with respect to one another, in order to arrange in a symmetric way (hexagonal configuration) the side electrodes, having a form of six posts electrically insulated from the other members. The aforementioned three points of support are in this case two contact points between the ball and two internal edges of the triangular opening, and the contact point between the ball and one of the side electrodes (see Figure 8b).

### 4.1. Non-Uniform Division of the Steregon

There are some applications where the full solid angle does not have to be divided into uniform detection sub-ranges. A few examples of respective tilt sensors are illustrated in Figure 9 (note that the lower and the upper electrodes are not visible). All the designs employ the aforementioned principle of three points of support and have a form of a regular prism. The following sensors feature respectively: 6 (Figure 9a), 12 (Figure 9b), and 16 (Figure 9c) detection sub-ranges.

The sensor presented in Figure 9a features the same six detection sub-ranges as the sensor presented in Figure 8. The difference is that the latter has the lower three detection sub-ranges rotated by 60 degrees arc with respect to the upper three sub-ranges.

All other kinds of prisms may be employed as the structure of a sensor based on three points of support of the ball: regular prisms with other types of bases or even oblique or truncated prisms.

An example of the first group is a tilt sensor with the structure of a right prism with a pentagonal base [48]. The sensor is presented in Figure 10. It has 7 electrodes and features 10 detection sub-ranges. The logic circuit dedicated to the sensor is an expanded version of the first circuit presented in Section 5, having one additional input and two additional outputs.

### 4.2. Uniform Division of the Steregon

In order to divide the steregon into uniform sub-ranges, it is advantageous to incorporate for this purpose Platonic solids, whose vertices create contact points with the circumscribed sphere; the points are centers of the detection sub-ranges [49]. However, there exist only five solids of this type: tetrahedron, cube, octahedron, dodecahedron, icosahedron—all presented in Figure 11.

From the technological point of view, only cube and dodecahedron are interesting, for a cube can be easily manufactured, whereas a dodecahedron provides the highest resolution of the sensor (it has 20 vertices). Tetrahedron has only four vertices (so, two times less than cube), while octahedron and icosahedron have four or five adjacent faces, respectively, creating one vertex (as a result, the free element is supported at three unknown points, what complicates the dedicated electronic circuit, which must be in such case a microprocessor system sequentially sampling the logic state of each electrode).

Examples of physical realization of a tilt sensor with cuboidal or dodecahedral structure are presented in Figure 12.

The cuboidal sensor has six electrodes, each perpendicular to each other; it features eight detection sub-ranges. The dodecahedral sensor has six pairs of opposite electrodes; it features 20 detection sub-ranges.

A more optimized design of the cuboidal sensor is proposed in Section 7, whereas a much simplified version, with regard to the shape of the casing, of the dodecahedral tilt sensor proposed in [50] is presented in Figure 13.

Principle of operation of the sensor is the same as previously: the ball is always supported on three electrodes, however the difference is that the electric contact is made either between the lower or the upper electrode and two adjacent side electrodes, or just between the lower and upper side electrodes, just like in Figure 13c (contact between the marked electrodes).

### 4.3. Shape of the Electrodes

If all the sub-ranges of the full solid angle are to be the same (for the sake of simplicity of interpretation), the contact surfaces of the electrodes must be either flat or convex. In the second case, position of the free element within a vertex of a solid is more stable (e.g., while subjected to vibration), yet at the cost of higher mechanical hysteresis. A higher mechanical hysteresis results in more extensive overlapping of the detection sub-ranges, which may be regarded as lower resolution of the sensor.

However, sometimes it would be advantageous to distinguish one of the sub-ranges. It is possible owing to creation of a concave shape of the contact surface of one of the electrodes. Then, it is possible to allocate a part of the related sub-range to a unique “zero position”, which can be interpreted as horizontal position (with certain precision). In such case, at small tilts, the free element is in contact with one bottom electrode only, and the sensor does not generate any signal, just like in the case of the sensors presented in Figure 4, Figure 5 and Figure 6.

All the possible shapes of the contact surface are presented in Figure 14.

It should be noted that it is possible to use various shapes of the contact surface of the electrodes at the same time. Whereas a concave shape of the contact surface may be applied to only one electrode (in order to keep the detection range full), the other shapes may be either flat or convex, without any limitations. In this way, it is possible to easily change the detection sub-ranges assigned to the triplets of particular electrodes.

## 5. Electronic Circuits

As previously mentioned, the biggest advantage of the considered type of tilt sensors is a possibility of operating without electric supply. Then, the appropriate adjacent electrodes, being electrically separated, simply break a dedicated circuit, which is closed only when the free member contacts the electrodes (two or three at the same time), i.e., only within a desired angular range specifying orientation of the sensor.

However, when the sensor is to generate a specific output signal (either analog or digital), the following options are possible:the logic state of particular electrode is changed,the total resistance of the sensor is changed, as proposed, e.g., in [4].

As far as the first option is concerned, an interesting solution created for tilt sensors having a prismatic shape is presented in Figure 15.

Terminals EL1 and EL6 are connected to the lower and the upper electrodes, whereas the other terminals (EL2–EL5) are connected to the side electrodes of the sensor. Each output (Y1–Y8) corresponds to a particular vertex of the prismatic sensor (only one output may be in high-level logic state at a time). Depending on the number of sides of the polygonal base of the prismatic sensor (see Figure 9), the circuit can be either reduced or expanded; addition (or removal) of any side wall requires addition (or removal) of one terminal (EL), one pair of logic gates (AND and NOR), four resistors, and two outputs (Y), since two new vertices are created.

The principle of operation of the circuit is the following. Making use of the fact that a right prism has two parallel bases, the lower is set to high-level logic state and the upper to low-level logic state. The free member (i.e., metallic ball) may be found only in one of the internal vertices of the prism, short-circuiting one of the bases with two adjacent side electrodes of the prism. Therefore, each orientation corresponds to a situation, when only two side electrodes are in high-level logic state (lower base down) or low-level logic state (upper base down). Each pair of adjacent side electrodes is connected to a pair of logic gates (AND and NOR), of which only one will have its output at high-level logic.

An example of the second option, where the total resistance of the sensor is changed due to short-circuiting of the electrodes by the metallic ball [50], is presented in Figure 16. A similar idea is proposed in [4].

However, if miniature size of the tilt sensor is strived for, another detection technique must be applied; then the mass of the free member is very low, so the contact resistance may be variable and extremely high—even in the order of MΩ. In such a case, the electrodes should be supplied with low alternative voltage (having the amplitude of a few Volts and the frequency of a few kHz). Under such a regime, proximity of the free member may be easily detected owing to a considerable reduction of the capacitance between the ball and the electrodes.

## 6. Experimental Studies

In order to find out what the mechanical hysteresis and repeatability of a sensor employing a ball as the free member is, appropriate experiments were conducted. Balls for rolling bearings, having various diameters (3.175–9 mm) were used.

The obtained results are presented in Table 1 [51]. The first raw concerns a ball with diameter of 5 mm tilted within a sensor of cuboidal structure, as in Figure 7. The second raw concerns the commercial tilt switch presented in Figure 2a, having the ball with diameter of 9 mm.

In the case of the first sensor, a lot of tests were carried out, with the ball located at each vertex of the cuboidal casing of the sensor. The sensor was both pitched and rolled. As far as the mechanical hysteresis and the repeatability are concerned, the tests proved that these parameters strongly depend on the ball diameter.

An additional test was carried out in order to determine the influence of roughness of the electrodes on the behavior of a ball with diameter of 3.175 mm. The ball was placed within a V-groove with very smooth surfaces and was tilted. In this case, the maximal value of the mechanical hysteresis was found to be of 9° [52].

In order to relate the performance of the solid tilt sensors to mercury sensors, other tests were carried out. Three mercury sensors with a cylindrical vial presented in Figure 1a were tested (the two sensors at the furthest left and the one at the furthest right). Results are presented in Table 2 [51].

To summarize, it can be stated that while keeping relatively large overall dimensions (i.e., over 10 mm) of the tilt sensor, whether it be a solid or a mercury sensor, the mechanical hysteresis may be smaller than 10° and the repeatability below 1°. However, while reducing the dimensions these parameters dramatically increase. Moreover, in the case of solid sensors, attention must be also paid to the roughness of the contact surface of the electrodes.

## 7. Manufacturing Issues

As far as conventional ways of manufacturing the considered sensors are concerned, their design should be adjusted accordingly. A good example may be the cuboidal sensor, originally presented in Figure 7 and Figure 12a. The cuboidal shape of the casing has been changed into a bush with two perpendicular ports, as illustrated in Figure 17. Another example is the dodecahedral sensor, presented in Figure 12b, whose optimized structure is illustrated in Figure 13.

An interesting alternative to the conventional ways of manufacturing the casing of the sensor is 3D printing, especially in view of satisfactory mechanical properties of some of the employed materials [52]. Then, complicity of the shape of the casing is usually of no concern.

As far as the free member is concerned, the most advantageous solution is to use balls for rolling bearings, since they feature very low surface roughness and low tolerance of the spherical profile. If the tilt sensor features only a single sensitive axis, the free member may be in the form of a roller instead of a ball, just like in the case of the tilt sensor presented in [4].

While striving for miniature dimensions of the sensor, the free member should be as small as possible. However, the smaller the diameter of the free member, the lower its mass (and thus the lower the unit pressure between the free member and the electrode) and the higher its resistance to motion. Therefore, the result is a higher transition resistance (which may be a problem while using measuring circuits like in Figure 15 and Figure 16) and a higher mechanical hysteresis. With regard to the former issue, a solution may be coating the electrodes and the free member with a noble metal (e.g., iridium, rhodium, platinum, gold), just like in the case of the switch presented in Figure 2a.

It is difficult to specify a size limit of the considered miniaturization. A certain problem is related to acquiring a miniature metallic ball with a smooth and hardened surface. As previously mentioned, ball bearings are a good source of such elements. The smallest diameter of this type of ball, known to the authors, is of 0.397 mm (1/64′′). Thus, a sensor of this type, featuring the overall dimensions of few mm can be easily fabricated.

## 8. Discussion

Since it is rather impossible to analyze all the existing tilt sensors, in Table 3 the most typical ranges of parameters are listed. All the considered types of sensors may feature dual-axis tilt detection over 360°, however it is a very difficult task in the case of mercury and liquid sensors. Due to the character of their operation, repeatability refers to mercury and mechanical solid sensors and switches, whereas sensitivity to liquid and MEMS sensors.

Since only one commercial solid switch was available, only single values of the hysteresis and the repeatability are given in the related row. Higher values of these parameters in the case of solid sensors result basically from lower diameter of the applied metallic ball (5 and 9 mm, respectively).

As far as commercial MEMS accelerometers are concerned, the lowest dimensions of their packaging are 1.1 × 1.1 × 0.74 mm, the negligible magnitude of mechanical hysteresis was reported in [53], whereas the ultra-high accuracy of 2′′ (0.0006°) was obtained for a pair of inclinometers operating in a differential configuration [54].

Analyzing the data reported in Table 3, the following can be stated:liquid sensors ensure the highest sensitivities, yet at the cost of large volumes;MEMS accelerometers ensure the smallest volumes and can operate under quasi-static conditions;mercury sensors feature the lowest measurement/detection range, yet they can carry large currents;solid sensors and MEMS accelerometers feature full measurement/detection range;it is relatively easy to change dimensions of solid sensors, depending basically on the diameter of the applied metallic ball; however, smaller diameters cause higher mechanical hysteresis.

It is difficult to provide detailed information on reliability and long-term stability of the analyzed types of sensors. However, it is well known that mercury sensors are very reliable and durable, for they contain no mechanical mating elements and, being hermetically closed, are not subjected to chemical reactions (provided the electrodes are made of a proper material and thus do not react with mercury). On the other hand, what may be surprising is that in the case of MEMS accelerometers, considerable aging effects were observed, especially in the case of older biaxial models: the resultant errors were evaluated at even 2.6% [55]. Moreover, problems with respect to their reliability after few years of operation were reported in the case of newer triaxial acceleration sensors [56].

As far as long-term stability of electric contact tilt sensors is concerned, it can be stated that while employing high quality metallic materials typically used for electric terminals (both for the electrodes and the ball as well as their coatings), and assuming a standard operation of the sensor, a high long-term stability can be expected. However, in case of operating the sensor under harsh mechanic or electric conditions (mechanical shocks and overloads, intense vibrations, large currents), or while employing untypical materials either for the electrodes or the ball, the long-term stability may be significantly lower. (Harsh chemical conditions should be eliminated by means of a hermitic housing or additional sealing.)

Under such circumstances, appropriate tests should be carried out in order to determine the real parameters pertaining to the long-term stability of the sensor, like, e.g., hardness, wear, short-circuit strength, or even mechanical strength—quoting an example of ceramic materials [57]. If the electrodes are made of elastic elements, as, e.g., in Figure 6, their fatigue strength should be tested as well.

In order to compare all the presented tilt sensors, their most distinctive features are listed in Table 4, including an approximate cost (assuming production on a large-scale). Depending on the nature of a given application, an appropriate sensor can be selected from among the designs presented in Table 4.

In order to increase reliability of electric contact tilt sensors (especially while miniaturizing their dimensions), the following detection technique should be used: supply with alternating voltage in order to eliminate the influence of variable electric resistance and the parasitic capacity between the ball and the electrodes.

## 9. Prospective Future Development

A very interesting example of recent related works may be a self-powered triboelectric sensor, using small balls with diameter of 6 mm, made of polytetrafluoroethylene [58]. Another noteworthy issue is the application of 3D printing technology for fabrication of a tilt sensor reported in [33]. These examples prove that new materials as well as new fabrication technologies provide means of developing new types of electric contact tilt sensors characterized by their typical advantageous features.

Aiming at further miniaturization, conventional steel balls used in bearings must be replaced with some other members. For example, using conductive microspheres (made of tin, silver, and copper alloy, which is usually used for soldering of ball-grid arrays) having diameter of 0.25 mm in the tilt sensor has already been successful, presented in [59]. Moreover, at the Institute of Micromechanics and Photonics, Warsaw University of Technology, a research project “Ultra-Efficient WIreless POwered Micro-robotic joint” (UWIPOM2) has been realized [60], which is aimed, among other things, at building a rolling bearing having external diameter of 300 µm and utilizing balls with diameter of 30–70 µm. Application of such micro-balls, featuring advantageous mechanical properties, makes it possible to build even smaller tilt sensors, having the overall dimensions of the order of single millimeters. Furthermore, such small balls may be applied to build a tilt switch having the structure analogical to the switch illustrated Figure 5; the advantage would be application of adhesion forces (of an appropriate value) between the ball and the casing instead of using a permanent magnet. In this way, not only miniature dimensions of the sensor may be obtained, but its simpler structure, and thus lower cost.

Even though there exist some original microsystem (MEMS) versions of tilt sensors, other than accelerometers, e.g., the capacitive sensor presented in [61], fabrication of a tilt sensor with a free member featuring diameter of single micrometers is rather out of question, since high adhesion forces in micro-scale [62], and even higher in nano-scale [63], would make the free member completely immovable.

Nevertheless, some kinds of tilt sensors, whose elements were successfully fabricated in micro- or nano-scale have already been reported. Their structure is completely different from the designs presented in this review: e.g., micropillars reported in [64], or nano-iron suspension discussed in [65]. It is highly probable that in the future, new types of micro- and nano-sensors similar to those presented in this review will appear.

## 10. Conclusions

Compared to application of low-*g* MEMS accelerometers for tilt measurements (which is a commonly accepted approach currently), the main shortcomings of electric-contact sensors are:bigger dimensions and mass,discrete operation only,low resolution,high mechanical hysteresis,operation under static conditions only, andhigher manufacturing cost at batch production (in the case of complicated designs).

However, advantages of the discussed electric-contact sensors still remain:no need for electric supply (while operating as a switching device),resistance to electrostatic discharges,simplicity of signal processing,possibility of shaping detection sub-ranges by means of appropriate design of the mechanical structure or the contact surface of the electrodes, andlow cost at piece production.

It seems that the most significant feature among the listed advantages is the capability of operation without electric supply. It is very important in the case of battery-supplied devices, which operate mostly in a sleeping mode, until they are triggered by some external mechanical stimulus that can be detected by a dedicated tilt switch. In this way, either their operation time can be considerably extended or the capacity of the battery decreased in a large measure, resulting in lower dimensions, smaller mass, and lower cost of the device.

Despite being a very old solution, there are some applications, where the aforementioned advantages of tilt sensors employing free members, remain unchallenged. The same refers to electrolytic liquid tilt sensors (inclinometers)—many types of such sensors are still manufactured, since they feature very high sensitivity, in the order of few arc seconds or even below arc second.

Moreover, dynamic development in material engineering and micro- and nanotechnology provides means for building electric contact tilt sensors with much smaller dimensions, simpler structure, and even new features.

## Figures and Tables

**Figure 1 sensors-21-01097-f001:**
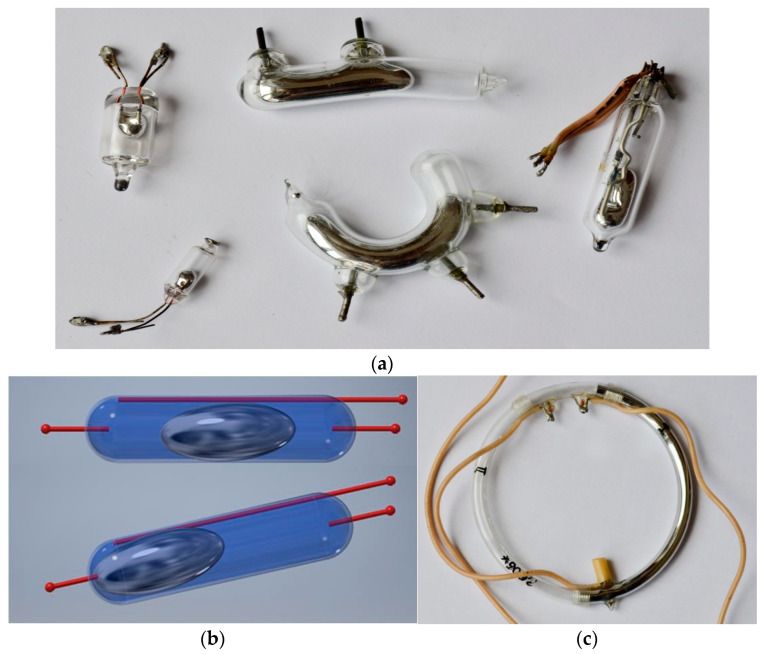
Mercury tilt sensors: (**a**) various types of contact switches; (**b**) principle of operation; (**c**) sensor with resistive wire.

**Figure 2 sensors-21-01097-f002:**
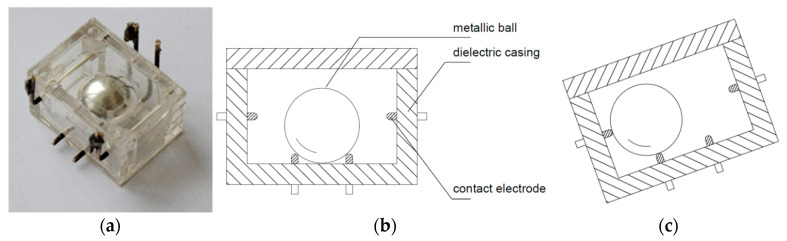
Single-axis tilt switch: (**a**) photograph; (**b**) general diagram; (**c**) principle of operation.

**Figure 3 sensors-21-01097-f003:**
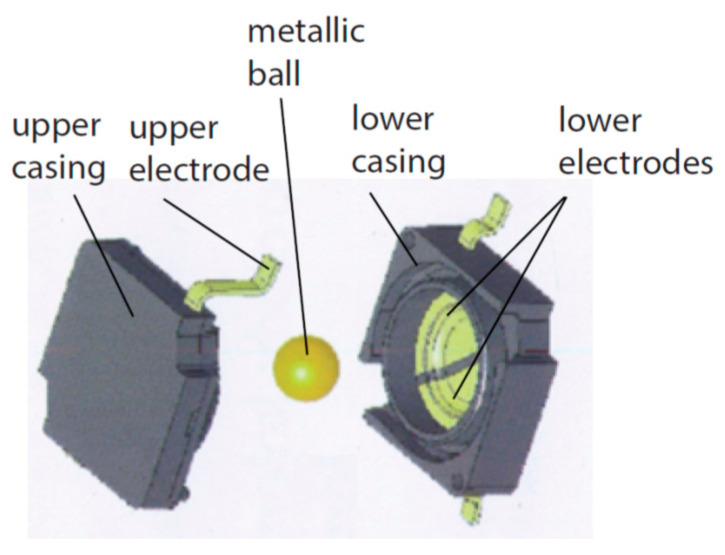
Dual-axis tilt switch by Alps Electric.

**Figure 4 sensors-21-01097-f004:**
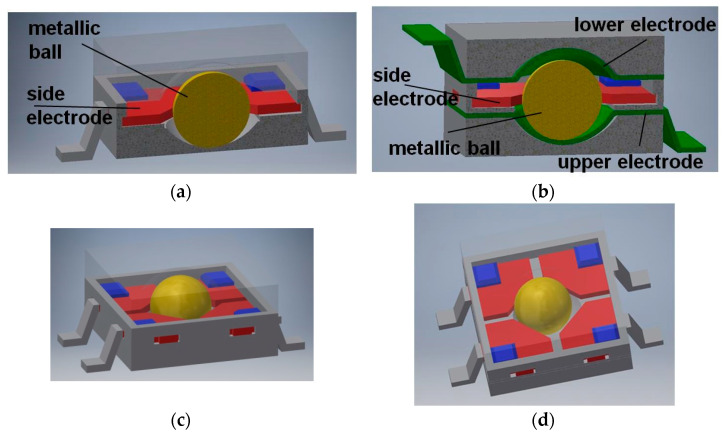
Dual-axis tilt switches: (**a**) with four electrodes; (**b**) with six electrodes; (**c**) arrangement of the side electrodes; (**d**) principle of operation.

**Figure 5 sensors-21-01097-f005:**
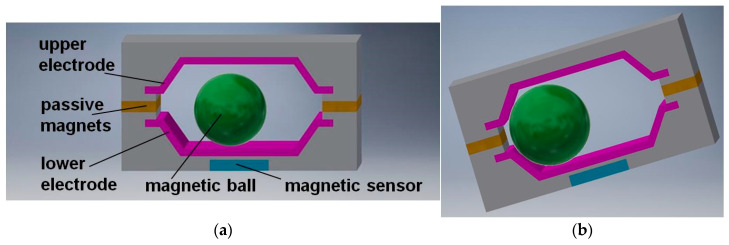
Tilt switch with motion detection: (**a**) the structure; (**b**) principle of operation.

**Figure 6 sensors-21-01097-f006:**
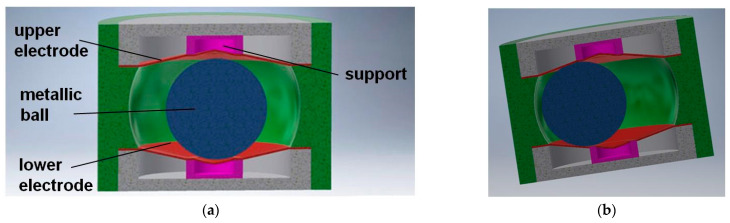
Dual-axis tilt switch with flexible electrodes: (**a**) the structure; (**b**) principle of operation.

**Figure 7 sensors-21-01097-f007:**
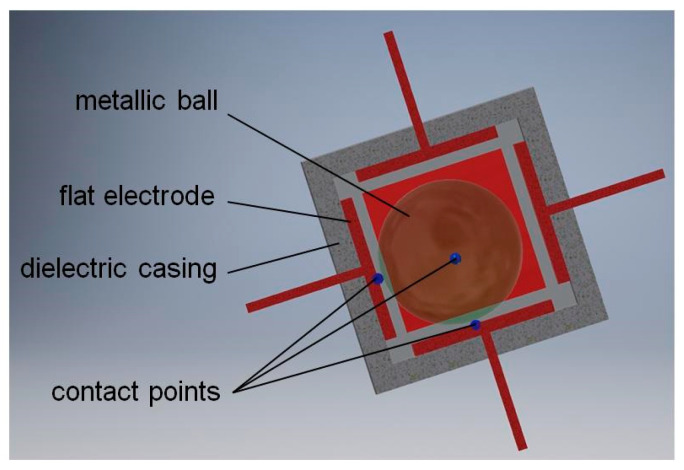
Principle of operation of a tilt sensor with eight detection sub-ranges.

**Figure 8 sensors-21-01097-f008:**
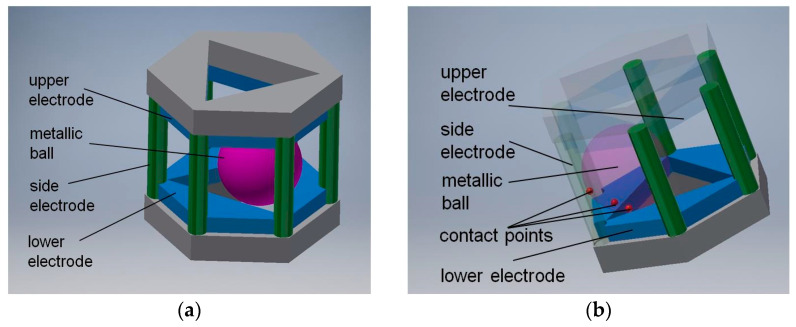
Sensor with six detection sub-ranges: (**a**) the structure; (**b**) principle of operation.

**Figure 9 sensors-21-01097-f009:**
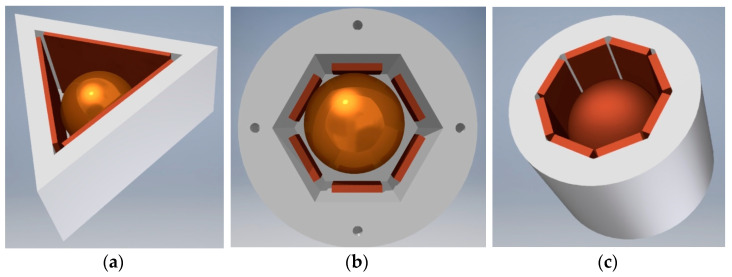
Tilt sensors with prismatic shape and spherical free element (view without the upper electrode): (**a**) triangular prism; (**b**) hexagonal prism; (**c**) octagonal prism.

**Figure 10 sensors-21-01097-f010:**
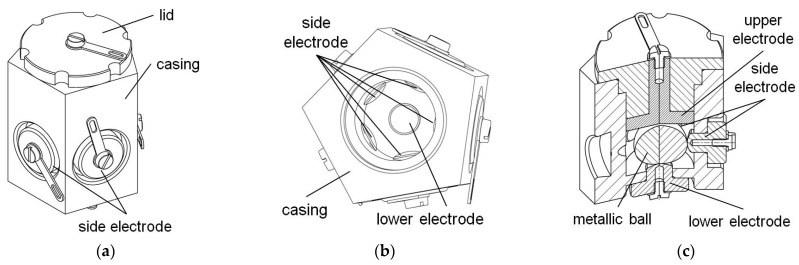
Tilt sensor in a form of pentagonal prism: (**a**) perspective view; (**b**) inside chamber; (**c**) establishing of electric contact (short-circuiting of the lower and the two side electrodes).

**Figure 11 sensors-21-01097-f011:**

The Platonic solids (number of vertices): (**a**) tetrahedron (4); (**b**) cube (8); (**c**) octahedron (6); (**d**) dodecahedron (20); (**e**) icosahedron (12).

**Figure 12 sensors-21-01097-f012:**
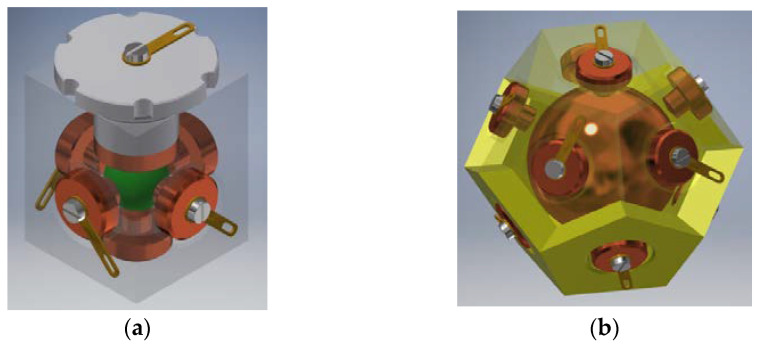
Tilt sensors based on the Platonic solids: (**a**) cuboidal sensor; (**b**) dodecahedral sensor.

**Figure 13 sensors-21-01097-f013:**
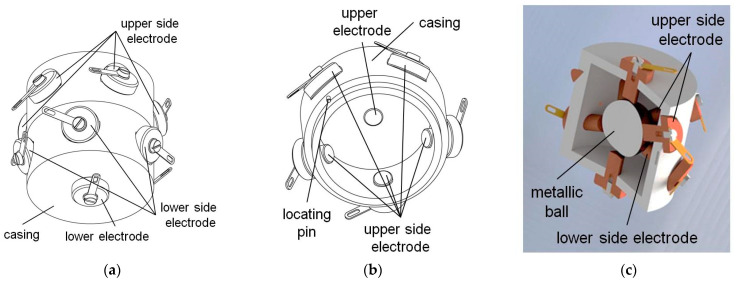
Dodecahedral tilt sensor: (**a**) perspective view; (**b**) inside chamber; (**c**) establishing of electric contact (short-circuiting of the upper side and two lower side electrodes).

**Figure 14 sensors-21-01097-f014:**
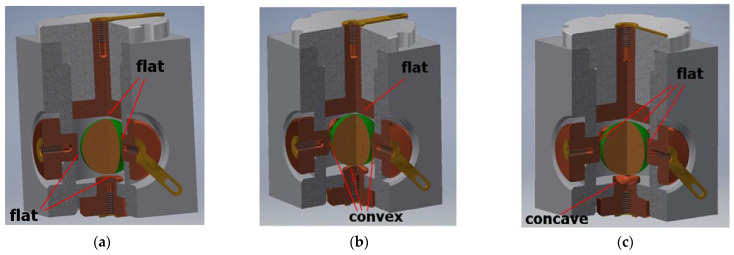
Possible shapes of the contact surface of the lower electrode: (**a**) flat; (**b**) flat and convex; (**c**) flat and concave.

**Figure 15 sensors-21-01097-f015:**
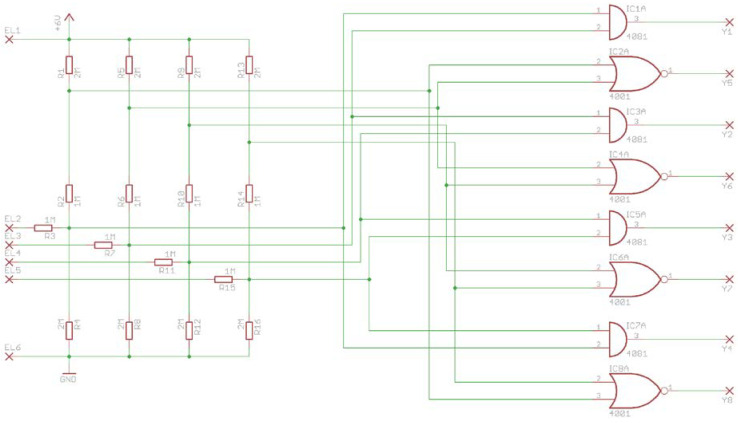
Logic circuit for the sensor in the form of the cuboidal sensor presented in Figure 12a: EL1–EL6, electrodes of the sensor; Y1–Y8, digital outputs corresponding to particular vertices.

**Figure 16 sensors-21-01097-f016:**
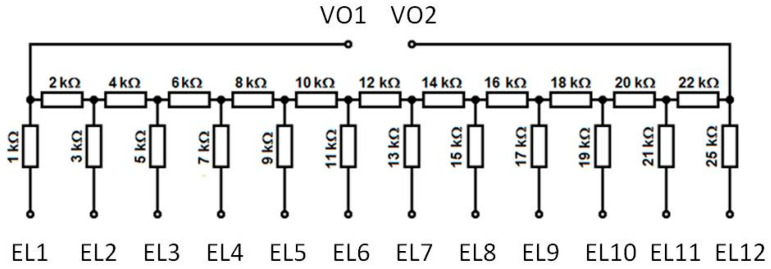
Resistance circuit created for the dodecahedron sensor illustrated in Figure 12b: EL1–EL12, electrodes of the sensor; VO1–VO2, analog voltage outputs.

**Figure 17 sensors-21-01097-f017:**
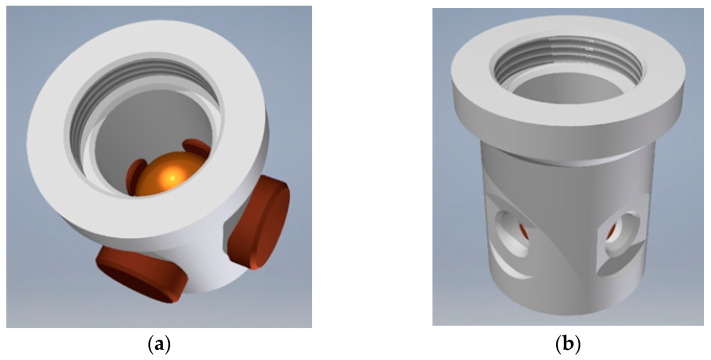
Cylindrical shape of the sensor illustrated in Figure 12a: (**a**) structural members; (**b**) shape of the casing (without the upper lid).

**Table 1 sensors-21-01097-t001:** Performance of the solid tilt sensors.

Sensor Volume	Mechanical Hysteresis	Repeatability
1.3 cm^3^	17°–48°	2.3°–11.9°
5.7 mm^3^	8.5°	1.2°

**Table 2 sensors-21-01097-t002:** Performance of the mercury tilt sensors.

Vial Volume	Mechanical Hysteresis	Repeatability
0.16 cm^3^	12.5°	0.9°
1.15 cm^3^	3.8°	0.3°
2.3 cm^3^	0.7°	0.2°

**Table 3 sensors-21-01097-t003:** Features of various tilt sensors.

Type of Sensor	Sensor Volume	Mechanical Hysteresis	Sensitivity/Repeatability	Operation Conditions	Measurement Range
Mercury	0.16–5.1 cm^3^	0.7°–12.5°	0.2°–0.9°	static	single axis, 30°
Liquid	1.3–31.6 cm^3^	–	0.0003°–0.05°	static	single axis, 180°
MEMS Accelerometer	0.0009–0.9 cm^3^	–	0.0006°–0.3°	quasi-static	dual axis, 360°
Solid Switch	0.03–5.7 cm^3^	8.5°	1.2°	static	single axis, 60°
Solid Sensor	0.03–10 cm^3^	17°–48°	2.3°–11.9°	static	dual axis, 360°

**Table 4 sensors-21-01097-t004:** Distinctive features of the presented tilt sensors.

Tilt Sensor	Type	Full Detection Range	No. of Detected Sub-Ranges	Special Features	Approximate Cost
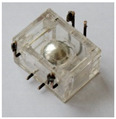	uniaxial	YES	4	good electric contact	$3
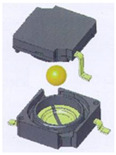	biaxial	NO	5		$5
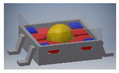	biaxial	NO	5		$5
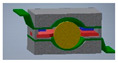	biaxial	NO	9		$6
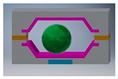	uniaxial	NO	3	detection of motion; smaller hysteresis	$10
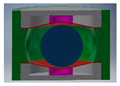	biaxial	YES	2	better electric contact; smaller ball	$10
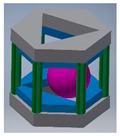	biaxial	YES	6	good electric contact	$25
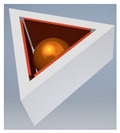	biaxial	YES	6 (7 *)		$20
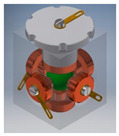	biaxial	YES	8 (9 *)	uniform sub-ranges	$20
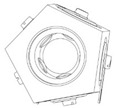	biaxial	YES	10 (11 *)		$20
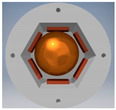	biaxial	YES	12 (13 *)		$20
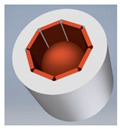	biaxial	YES	16 (17 *)		$20
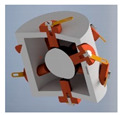	biaxial	YES	20 (21 *)	uniform sub-ranges	$30

* higher number of sub-ranges is possible in the case of applying a concave shape of one of the electrodes (see Section 4.3).

## Data Availability

Not applicable.

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
