# Peer review of "Electric-Contact Tilt Sensors: A Review"

_sensors, 2021, doi:10.3390/s21041097_

Round 1

Reviewer 1 Report

The paper is revised and improved to a certain level.

Author Response

Dear Reviewer,

Thank you for the positive evaluation of the revised version.

Reviewer 2 Report

  1. This paper presents a review of various kinds of solid tilts sensors, which provides good reference in this field.
  2. The paper is written well; only some minor corrections are recommended, as listed below.
  3. In Line 44, “MEMS”, please use the full name for the first time. This applies to other places such as in Line 55, “ABS”, etc.
  4. Line 134, “switch by…” should delete “by”
  5. Lines 169 and 171, what’s the meaning of 5 and 4 in the text?
  6. Lines 272 and 273, 4a-4e, 5a-5e should be marked in Fig. 13.
  7. Line 465, “Compare to…” should be “Compared to…”
  8. Reference 64 is not used in the text.

Author Response

Dear Reviewer,

Thank you for your vigilance.

Point 1, Point 2: Thank you for the positive evaluation of the revised version.

Point 3: The abbreviations have been explained.

Point 4: The sentence has been corrected.

Point 5: The numbers have been deleted.

Point 6: The numbers have been deleted.

Point 7: The sentence has been corrected.

Point 8: Thank you for your vigilance. There was an error with numbering of the references; it has been fixed.

Reviewer 3 Report

First of all, I have to disclose that I mainly work with more sensitive tiltmeters than those discussed in this article. Therefore, my comments may be biased. This manuscript reviews the mechanics of electric-contact tilt sensors. I feel that this manuscript is well written as a review of electric-contact tilt sensors. Therefore, this article merits publication. However, I have a few comments which I believe strengthen the manuscript. 1. Why do the authors make the review exclusive to electric-contact sensors? Tiltmeters with other mechanics are available; for example, tiltmeters to measure subtle (~10 nanoradians) changes use various kinds of pendulums. I would not suggest the authors cover all the spectrum of tilt sensors, but I want the authors to explain the motivation for exclusively reviewing electric-contact sensors. 2. I found that the review of sensors of various mechanics is well written, but I did not understand the differences among them. In other words, I want to understand the pros and cons of each sensor. A table summarizing them should help. 3. As electric-contact sensors are not as sensitive as the most sensitive sensors to be employed in, for example, geophysical and astronomical studies. Therefore, the cost is an essential factor in selecting which sensors to choose. I want to see arguments on the cost and benefit of each type of sensor. 4. Are the sensors reviewed in this article for continuous monitoring? In that case, temporal stability is also essential. I want to see an argument of temporal stability if these sensors are used for continuous monitoring of an object.

Author Response

Dear Reviewer,

Thank you for your valuable comments. Let us briefly address your points:

Point 1: We have limited the review basically to electric-contact sensors since they can be manufactured as miniature devices, with dimension in the order of tens of millimeters. Please note that the manuscript has been submitted to the Special Issue: "Special Applications of microsensors". Besides, we did not want to increase volume of the review.

Point 2: Following your apt suggestion, we have added Table 4: "Distinctive features of the presented tilt sensors", where their most important differences are discussed. Besides, in section 10 we have added a paragraph related to the capability of operating without electric supply.

Point 3: Following your apt remark, we have briefly addressed the issue of cost in the new Table 4.

Point 4: Following your apt remark, we have briefly addressed the issue of temporal stability in a new paragraph started in line 436.

Reviewer 4 Report

It is a very simple and instructive paper. It is very well laid out, written in a way that is easy to read.

Must be Add :

Line 46: indicate what MEMS means

Line 55: indicate what ABS means

Author Response

Dear Reviewer,

Thank you for your vigilance.

The abbreviations have been explained.

This manuscript is a resubmission of an earlier submission. The following is a list of the peer review reports and author responses from that submission.

Round 1

Reviewer 1 Report

The paper discusses "Electric-contact tilt sensors" from various kinds of view points. Thus, I have some comments and questions.

The paper discusses "Electric-contact tilt sensors" from various kinds of view points. Thus, I have some comments and questions.

* Figure 1,3,4,5,6,7,8,11,12,15: You should put some words indicating some parts in the figures for easy understanding the figures.

*Figure 1: There are many kinds of mercury tilt sensors. You should add an illustration for explaining the working principle.

*Figure 2,3,4,- - -:These are contact tilt sensors. You shoud add an illustration for explaning the working principle.

*Figure4: You should put the words such as "4 electrode", "6 electrode" and "side electrodoes" in the figure for easy understanding the figure.

*Figure 5: You should add an illustration for explaining the working principle.

*Figure 6:  You should add an illustration for explaining the working principle.

*Figure 7: You should put the word, "8 detection parts" in the figure for easy understanding it.

*Figure 8: You should add an illustration for explaining the working principle.

*Figure 12: you should indicate "flat", "convex" and "concave" in the figure for easy understanding the figure.

*Figure 12: The figure caption should unites with the figure.

*Figure 13: You should add some words such as "Input Vi1, Vi2, etc.", "Output Vo1, Vo2, etc." in the figure for easy understanding the circuit.

*Chapter "9 Patents": I think that chapter "9 Patent" is not appropriate for the journal paper.

*I like to know the accuracy of the cuboidal sensors compared with conventonal tilt contact sensors.

*I like to know commercial tilt sensors compaperd with the new tile sensors shown in the paper.

Author Response

Thank you very much for your apt and thoughtful remarks and comments, owing to which we were able to considerably improve our manuscript.

First, we have revised 4 pictures (Fig. 3, 7, 14, 16) and added 7 new pictures (Fig. 1b, 2b, 2c, 4d, 5b, 6b, 8b), which explain the operation principle of particular sensors, as you have suggested.

Except for Fig. 14 (Fig. 12 previously), we did not follow your advice of putting the descriptive words in the figures, even though that would surely improve their legibility. The reason for such decision is that we did not want to increase volume of the manuscript (19 pages, large numbers of pictures: over 40 altogether, 15 with labels). However, we expanded captions of some figures (Fig. 3, 7, 8, 14, 15, 16), for a better understanding.

For aesthetic reasons, we did not change symbols in the diagram in Fig. 15, as you suggested, yet we did it in the diagram in Fig. 16; besides, we added a uniform explanation to the captions of these figures.

We followed your advice, and shifted the content of Section 9 (Patents) to Section 4.1 and 4.2, respectively.

With regard to the accuracy of the new sensors compared with conventional tilt contact sensors, we provided the only data we have in Table 1 (they refer to only one piece of each type - see explanation in lines 426-428). However, since the mechanical operation principle is always the same, the presented values are fully representative. At the time being (having only a limited access to the laboratory) we are not able to purchase and test some other commercial switches (accuracy/repeatability of such sensors is usually not specified in the related datasheets). Besides,  only a prototype of the cubic sensor (Fig. 12a) has been manufactured so far. Designs presented in Fig. 10 and 13 have not been prototyped yet. Nevertheless, in Conclusions we added Table 3, comparing concisely the most typical tilt sensing techniques.

Reviewer 2 Report

This submission presents an overview of tilt sensor approaches.  Whilst this reviewer cannot claim to be an expert in tilt sensor technology – there is here considerable general interest in sensing approaches coupled to detailed background and much experience in a few technological sensing disciplines.  Since the review should interest the general reader rather than the expert, then maybe this is appropriate.

In its present form, the review is little more than a list of commercially available techniques.  Whilst there is some mention of some performance parameters and application sectors, there remains a significant need for a concise presentation of comparative benefits and disadvantages of each approach based on individual performance specifications.  In most technology reviews of this nature, this is conveniently approached as a table in which critical parameters (sensitivity, hysteresis, long term reliability, ability to withstand environmental specifications – temperature variations, vibration at the like - may be among the parameters to consider) are listed and a ‘comments’ column highlights interesting benefits and limitations.  There is certainly scope for such treatment here – the present manuscript needs concise critical assessments of the technologies presented.  There is also a perhaps too frequent mention of references 40 and 41 - not to mention an appendix (in effect – it follows the conclusions) which mentions some key features.  But what is really needed here is – what motivated the approach to this design and what and where are the user benefits?

There are also a number of, possibly typographical, possibly linguistic, errors throughout the manuscript which need correction (one example – just after figure one, ‘emerged’ rather than ‘immersed’)

But most of all, for this reviewer a concise critical assessment of all the technologies mentioned and their relative merits and disadvantages is an essential feature of any review manuscript.  This is currently lacking.  Additionally, the authors need to reconsider the multiple mentions of their patent especially since the potential benefits have not yet been described and as far as this reader can tell, the thorough experimental verifications necessary to establish its potential prospects and the essential cost benefit analysis are currently, at best, in progress.  However, the concept of publishing the review certainly has merit and could be of significant interest.  The intention in the review is to assist in the process of enhancing the presentation such as to maximise its possible future impact. contribute

Author Response

Thank you very much for your apt and thoughtful remarks and comments, owing to which we were able to considerably improve our manuscript by adding a missing content.

Following your advice, we considerably extended the Conclusions. Table 3 was added, comparing concisely the most typical tilt sensing techniques. Based on their advantages and shortcomings, we specified application scope of each technique. We added some more features in Tab. 3 (compare to Fig. 1 and 2). In order to do it, 2 new references (research articles [46]-[47]) were added. However, we were not able to specify all the parameters you mentioned due to lack of specific data. We only referred to reliability and durability of mercury sensors in a general way, and to our own study referring to long-term stability of MEMS accelerometers, reported in other 2 added references (research articles [48]-[49]) - see the paragraph in lines: 443-450.

As you suggested, the frequent mention of references 40 and 41 (now: 39 and 41) was reduced to 3 times.

With regard to "motivation of the approach to this design and what and where are the user benefits", we added an extended explanation in lines 186-189, pointing to simplicity of processing and analyzing the output signals.

We corrected the error regarding ‘emerged’/‘immersed’, along with many other lingual errors - thank you very much for a close reading of the text.

Reviewer 3 Report

Major revision of the text is needed especially the introduction which is not an introduction at all according to normal standards.

The introduction is a juxtaposition of opinions by the authors but sometime no relation at all with the topic in hand (biomimetic consideration of the inner ear of animals), or opinions not substantiated or evidenced by the authors (MEMS provide better tilt information with accelerometers). The introduction also does not present the organisation of the article at all. 

There is therefore a major revision of the article to be done just for this part even before attempting to read the rest of the article.

Please read how other articles prepare their presentation and emulate them.

Author Response

Thank you very much for your apt and thoughtful remarks and comments, owing to which we were able to considerably improve our manuscript.

With regard to the introduction. We understand that you expected a more concise and specific content. However, since the considered tilt sensors are quite simple devices, we decided to maintain a more general character of the Introduction, especially in view of the fact that MEMS accelerometers and liquid sensors are applied in tilt measurements more often than the contact sensors/switches. Thus we also considerably extended Conclusions, comparing the most typical tilt sensing techniques and specifying application scope of each one of them.

Nevertheless, we slightly shortened the fragment related to the analogy of the sense of balance in the case of mammals and fish (please note that fish additionally sense the gravity and sometimes use it for navigation purposes). We added a paragraph related strictly to the considered sensors (lines 53-60),

Following your suggestion, we added information about the organization of the text (which has been changed: Section 9 (Patents) has been shifted to Section 4.1 and 4.2) - (lines 61-67).

Round 2

Reviewer 1 Report

(1)Figure 1: You should put some words indicating some parts in the figures (a), (b) and (c) for easy understanding the figures. Is Figure 1(b) corrected?

(2)8.Conclusins: This part is some kind of "Discussions" NOT "Conclusions". I think that the revision is necessary.

Reviewer 2 Report

The authors have done a very effective job in re-organising the manuscript.  It is now ready for publication.  In particular the conclusions are now much more informative.

Reviewer 3 Report

It is disappointing that little efforts have bene made to enhance the quality of the article.

A fundamental revision of the introduction was needed.

I suggest that the authors do follow a course on how to write scientific articles as there seems to be some misunderstanding as to how a journal article should be put together.